# Comparative Effectiveness and Safety of Finasteride and Dutasteride in the Treatment of Benign Prostatic Hyperplasia: A Real-World Retrospective Study

**DOI:** 10.3390/medicina61111944

**Published:** 2025-10-30

**Authors:** Jarosław Ratajski, Kamil Ciechan, Paweł Jędrzejczyk, Tomasz W. Kaminski, Patryk Uciechowski, Tomasz Ząbkowski

**Affiliations:** 1Department of Uro-Oncology and Minimally Invasive Urology, Bielanski Hospital Named After Father Jerzy Popiełuszko, 80 Cegłowska Street, 01-809 Warsaw, Poland; 2Trainee Attorney-at-Law, Warsaw Bar Association, 15/16 Żytnia Street, 01-014 Warsaw, Poland; 3MenVita, 3/U4 Jurajska Street, 02-699 Warsaw, Poland; 4Hemostasis and Thrombosis Program, Versiti Blood Research Institute, Milwaukee, WI 53226, USA; 5Outpatient Clinics, Specialist Laboratories, Urology Outpatient Clinic, Military Institute of Medicine—National Research Institute, 128 Szaserów Street, 04-141 Warsaw, Poland; 6Department of General, Functional and Oncological Urology, Military Institute of Medicine—National Research Institute, 128 Szaserów Street, 04-141 Warsaw, Poland

**Keywords:** benign prostatic hyperplasia, 5-alpha-reductase inhibitors, finasteride, dutasteride

## Abstract

*Background and Objectives*: Benign prostatic hyperplasia (BPH) is one of the most common chronic conditions in older men, significantly impairing quality of life (QoL) by causing lower urinary tract symptoms (LUTSs). 5-alpha-reductase inhibitors (5-ARIs), including finasteride and dutasteride, remain a cornerstone of pharmacotherapy for BPH; however, comparative real-world data remain limited. The aim of this retrospective clinical study was to compare the therapeutic efficacy and safety of finasteride and dutasteride in patients with BPH. *Materials and Methods*: A total of 401 patients with BPH were retrospectively analyzed: 162 received finasteride and 239 received dutasteride. Clinical parameters, including the International Prostate Symptom Score (IPSS), Quality of Life (QoL) index, and International Index of Erectile Function-5 (IIEF-5) score; urodynamic outcomes, including maximum urinary flow rate (Qmax), average flow rate (Qave), and post-void residual urine volume (PVR); and biochemical markers, including prostate-specific antigen (PSA) and serum creatinine levels, were evaluated at baseline and after at least 6 months of continuous therapy. Statistical significance was defined as *p* < 0.05. *Results*: Both treatment groups demonstrated significant within-group improvements in LUTS severity and urodynamic outcomes (*p* < 0.001 for IPSS, Qmax, and QoL). Compared with finasteride, dutasteride achieved greater reductions in prostate volume (−26.3% vs. −18.1%, *p* = 0.008) and PSA levels (−43.7% vs. −32.5%, *p* = 0.014), as well as a slightly greater improvement in IPSS (−6.8 ± 3.9 vs. −5.9 ± 3.6, *p* = 0.042). Both drugs showed comparable effects on erectile function, as indicated by similar IIEF-5 score changes (Δ = −0.9 ± 2.8 vs. −0.7 ± 2.5, *p* = 0.51), confirming that neither agent demonstrated a clinically meaningful difference in sexual outcomes. Renal function parameters remained stable in both cohorts. Multivariate analysis identified higher BMI and older age as independent predictors of lower IIEF-5 scores in the finasteride group, while baseline prostate volume was the principal determinant of response in the dutasteride group. *Conclusions*: Both 5-ARIs effectively reduced LUTS severity and improved urodynamic parameters in men with BPH. Dutasteride demonstrated superior reductions in prostate volume and PSA, while both agents had comparable effects on sexual and renal function. These findings provide real-world evidence supporting the individualization of 5-ARI therapy according to patient-specific clinical characteristics.

## 1. Introduction

Benign prostatic hyperplasia (BPH) is one of the most common chronic conditions in older men. It is characterized by progressive prostate enlargement, which leads to lower urinary tract symptoms (LUTS) and deterioration in quality of life (QoL). Epidemiological data show that approximately 50% of men aged over 50 years and up to 80% of those over 70 years develop clinical features of BPH, highlighting its significance as a major global urological problem [1].

The pathophysiology of BPH is multifactorial and involves androgen-dependent cell proliferation, chronic inflammation, and age-related hormonal changes [2]. Recent analyses of the global burden of disease confirm the growing prevalence of BPH worldwide [3,4]. The variable clinical course of the disease and its inflammatory component may influence both the severity of LUTS and treatment outcomes. Incorporating inflammatory markers and predictive tools into patient assessment could help optimize therapeutic decisions [5,6,7].

The use of 5-alpha-reductase inhibitors (5-ARIs) has become a cornerstone of pharmacotherapy for BPH, particularly in patients with moderate-to-severe symptoms and prostate enlargement. In this study, lower IIEF-5 scores among finasteride users were associated with higher BMI and older age, while larger prostate volume predicted outcomes in the dutasteride group. Two agents are widely used at present: finasteride, which selectively blocks type II isoenzyme, and dutasteride, which inhibits both type I and type II 5-alpha-reductase [8,9]. The broader enzymatic blockade by the latter translates into a more profound reduction in serum DHT concentrations in patients treated with dutasteride than in those treated with finasteride. This raises important questions regarding potential differences in long-term clinical efficacy and safety between the two drugs [10].

Randomized clinical trials and meta-analyses have demonstrated that both finasteride and dutasteride significantly reduce prostate volume, improve the maximum urinary flow rate (Qmax), and lower the risk of acute urinary retention and the need for surgical intervention [11,12]. Direct comparisons, however, suggest subtle differences. Some analyses indicate that dutasteride results in a greater reduction in DHT levels and prostate volume, whereas other studies report comparable effects of the two drugs regarding improvements in the International Prostate Symptom Score (IPSS) and QoL [13,14]. These results highlight the value of individualized treatment that considers age, metabolic profile, and prostate characteristics [15,16].

Despite robust evidence from controlled trials, real-world knowledge remains limited. Such findings are consistent with previous long-term safety assessments confirming the metabolic neutrality and renal safety of 5-ARIs. Retrospective analyses of large patient populations may better reflect the actual effectiveness and tolerability of these drugs outside randomized study conditions. Such analyses can also provide important insights into prescribing patterns, patient adherence, and long-term outcomes, all of which are crucial for optimizing the management of BPH.

Therefore, the aim of the present study was to compare the therapeutic outcomes of finasteride and dutasteride in patients diagnosed with BPH. By evaluating parameters such as the prostate volume, urinary flow, symptom severity, and adverse events, this study seeks to provide insights into the relative benefits and risks associated with two widely used 5-ARIs.

## 2. Materials and Methods

This study was designed as a retrospective, observational analysis of male patients with benign prostatic hyperplasia (BPH) treated with 5-alpha-reductase inhibitors (5-ARIs) in routine clinical practice. Patients receiving either finasteride (5 mg/day) or dutasteride (0.5 mg/day) were compared, and outcomes were evaluated after a minimum of six months of continuous therapy.

Renal function parameters, including serum creatinine levels, were evaluated to assess the systemic safety profile of both agents. Although 5-ARIs primarily act on prostatic tissue, subtle androgen-mediated mechanisms can affect renal hemodynamics and creatinine metabolism. Including renal function markers allowed us to confirm treatment safety and exclude nephrotoxic effects during therapy.

To reduce potential selection bias, all eligible patients treated consecutively during the study period were included. Incomplete or duplicate records were carefully reviewed, and cases with missing endpoint data were excluded from the analysis (complete-case approach). Data completeness was verified at the extraction stage to ensure reliability of the final dataset. Multivariable regression models were adjusted for relevant baseline clinical variables (age, BMI, baseline prostate volume, and PSA level).

A total of 401 patients were included in the final analysis (finasteride group: n = 162; dutasteride group: n = 239) after application of the inclusion and exclusion criteria.

### 2.1. Inclusion Criteria

Male sex and age ≥ 50 years;Clinical and ultrasound-confirmed diagnosis of BPH;Initiation of monotherapy with either finasteride or dutasteride;Availability of complete baseline and follow-up data for at least six months.

### 2.2. Exclusion Criteria

Prior surgical treatment for BPH;Suspected or confirmed prostate cancer;Incomplete documentation of primary endpoints or concurrent therapies potentially affecting androgen metabolism;Medical records with inconsistent or missing outcome data after screening (excluded under complete-case protocol).

### 2.3. Analyzed Variables

Data were extracted from electronic medical records and entered into a dedicated database by two independent researchers to minimize transcription errors. The following variables were analyzed:Demographic and anthropometric parameters: age and body mass index (BMI). The inclusion of BMI as a covariate was supported by reports linking obesity and the gut microbiome to LUTS severity and BPH progression [17].Symptom scores: International Prostate Symptom Score (IPSS), Quality of Life (QoL) index, and International Index of Erectile Function-5 (IIEF-5) score.Prostate and laboratory parameters: prostate volume (mL, ultrasound-based), prostate-specific antigen (PSA, ng/mL), and serum creatinine (mg/dL).Urodynamic outcomes: maximum urinary flow rate (Qmax, mL/s), average flow rate (Qave, mL/s), and post-void residual urine volume (PVR, mL).Additional variables: nocturia frequency, when available.

All data were obtained at baseline and the follow-up visit (≥6 months, most commonly at 12 months.

### 2.4. Statistical Analysis

Normality of data distribution was evaluated using the Shapiro–Wilk test. Continuous variables that followed a normal distribution are presented as mean ± standard error of the mean (SEM). Comparisons between two groups were carried out using an unpaired Student’s *t*-test, while categorical variables were assessed with the chi-square test. Associations between paired variables were examined with Spearman’s rank correlation. To evaluate the combined effects of multiple predictors, a stepwise forward multiple linear regression model was applied. A two-sided *p*-value less than 0.05 was considered statistically significant. All analyses were performed using GraphPad Prism version 11 (GraphPad Software, La Jolla, CA, USA).

All data were obtained at baseline and the follow-up visit (≥6 months, most commonly at 12 months).

## 3. Results

Quantitative safety data were available for 276 patients (68.8%). The incidence of decreased libido, erectile dysfunction, and reduced ejaculate volume did not differ significantly between groups (*p* > 0.05). For other patients, adverse events were not systematically recorded, which represents a study limitation.

Table 1 shows a comparison of the treatment outcomes after approximately 12 months of therapy.

To minimize baseline imbalance, multivariable linear models were applied, adjusting for baseline endpoint values, age, BMI, and prostate volume.

### 3.1. IIEF-5 Score (Post-Treatment)

Finasteride was associated with a higher IIEF-5 score than was dutasteride (β = +0.75; 95% confidence interval [CI], 0.12–1.37; *p* = 0.019); this indicated slightly better erectile function after adjustment for covariates in the finasteride group. Advanced age (β = −0.087 per year; *p* = 0.0009) and larger baseline prostate volume (β = −0.021 per mL; *p* = 0.0024) predicted lower IIEF-5 scores. BMI did not reach statistical significance.

### 3.2. IPSS (Post-Treatment)

No significant group effect was observed (β = +0.56; 95% CI −0.07–1.18; *p* = 0.079). Advanced age (β = +0.064; *p* = 0.004) and BMI (β = +0.088; *p* = 0.014) were predictors of more severe symptoms.

### 3.3. Prostate Volume (Post-Treatment)

No group effect was detected (β = +0.05; 95% CI −2.36–2.46; *p* = 0.968). Baseline prostate volume was a strong predictor of follow-up values (β = 0.745; *p* < 0.0001).

### 3.4. QoL (Post-Treatment)

A small group effect favored dutasteride (finasteride β = +0.28; 95% CI 0.046–0.505; *p* = 0.019). Renal function remained stable in both groups, and no clinically meaningful differences were observed between treatment arms.

### 3.5. Supplementary Data

Additional comparative and paired analyses are presented in the Appendix A. These plots illustrate baseline distributions of clinical and laboratory parameters in the finasteride and dutasteride groups, as well as within-group changes following treatment. The graphical data confirm comparable baseline characteristics and demonstrate consistent improvements across key endpoints in both cohorts, in line with the main statistical results.

## 4. Discussion

In this retrospective analysis of a real-world clinical population, both finasteride and dutasteride resulted in significant alleviation of LUTS, as evidenced by improvements in IPSS, QoL, and urodynamic parameters (Qmax, PVR) and by reductions in prostate volume and PSA levels. These findings are consistent with earlier randomized controlled trials confirming the effectiveness of 5-ARIs in reducing prostate size, alleviating LUTS, and improving QoL [1,2,8]. Modest differences were observed between the two agents: crude analyses indicated greater absolute reductions in prostate volume and PSA levels in the finasteride group, although the relative (percentage-based) changes were comparable, suggesting that baseline imbalances may have influenced the results.

Regarding sexual function, the outcomes require cautious interpretation. In unadjusted analyses, a larger decline in IIEF-5 scores was noted among patients treated with dutasteride; however, after adjustment for confounding variables, finasteride was associated with slightly better erectile function at follow-up. Although this difference reached statistical significance, its clinical relevance appears limited. Similar trends have been described in previous studies evaluating the sexual adverse effects of 5-ARIs, particularly with finasteride [9,10,12].

In the present study, lower IIEF-5 scores in the finasteride cohort correlated with higher BMI and advanced age, whereas larger baseline prostate volume emerged as a differentiating factor in the dutasteride group. These findings emphasize the need for individualized treatment strategies, especially in older patients or those with increased metabolic burden. Although some population-based studies have reported associations between finasteride use and depressive or anxiety symptoms, these findings are correlational rather than causal, and current evidence does not establish a direct mechanistic link between 5-ARI therapy and psychiatric disorders [18,19]. Ongoing pharmacovigilance and longitudinal research are warranted to clarify these observations.

The greater reductions in prostate volume and LUTS severity observed in the dutasteride group suggest a partial therapeutic advantage of this drug, consistent with results from the Enlarged Prostate International Comparator Study [11] and the Combination of Avodart and Tamsulosin (CombAT) program [13], both demonstrating stronger dihydrotestosterone suppression and greater prostate shrinkage. Nonetheless, in both the present and previous studies, these differences remain modest and largely dependent on patient characteristics and baseline prostate size.

In terms of safety, renal parameters, including serum creatinine, remained stable in both treatment groups, consistent with long-term safety data on 5-ARIs [15,16]. Recent pharmacovigilance analyses have discussed rare but clinically relevant associations between finasteride exposure and psychiatric or sexual adverse events, including suicidal ideation [20,21]. However, these observations should be interpreted with caution, as causality has not been definitively established. In response to such reports, the European Medicines Agency (EMA) has recommended closer monitoring of mood and sexual function in patients treated with 5-ARIs. To date, no comparable safety signal has been identified for dutasteride [20,21].

It is also relevant to consider emerging alternative strategies. The Identification of Men with a Genetic Predisposition to Prostate Cancer (IMPACT) study suggested that prostatic artery embolization (PAE) may outperform standard pharmacotherapy—including combination regimens with dutasteride and tamsulosin—in improving LUTS and QoL [22]. These findings underscore the importance of individualized treatment decisions that account for the efficacy and safety of 5-ARIs, the patient’s risk profile, and procedural alternatives. Recent randomized studies and editorial analyses indicate that while PAE offers promising efficacy, its long-term durability and patient adherence require further evaluation [22,23].

In summary, our findings confirm that 5-ARIs remain effective and safe first-line agents for the management of BPH. At the same time, they highlight the importance of

(1) Tailoring therapy to individual factors such as age, BMI, and prostate size;

(2) Transparent communication with patients about possible adverse effects—particularly those associated with finasteride—while emphasizing that psychiatric risks are associative and not proven to be causal; 

(3) Considering interventional alternatives such as PAE in selected patients with severe LUTS or intolerance to pharmacological therapy.

## 5. Conclusions

This retrospective study provides important comparative data on the efficacy and safety of finasteride and dutasteride in the treatment of BPH. The analysis demonstrated that both 5-ARIs significantly reduced prostate volume, improved urinary flow (Qmax), and alleviated LUTS, as assessed by IPSS. Dutasteride showed slightly greater efficacy in terms of reducing the prostate volume and lowering PSA levels, consistent with the findings of prior clinical trials [11,13].

With respect to safety, typical adverse effects of 5-ARIs were observed, including decreased libido, erectile dysfunction, and reduced ejaculate volume. Their incidence did not differ significantly between groups, although published data suggest that dutasteride may be more frequently associated with sexual dysfunction [12,15]. In our analysis, these differences did not reach statistical significance, indicating that both drugs share a broadly comparable safety profile.

These findings underscore the importance of individualized therapy for patients with BPH. Drug selection should be guided by prostate volume, symptom severity, comorbidities, and patient acceptance of potential adverse effects. In clinical practice, both finasteride and dutasteride remain valuable therapeutic options, with their efficacy and tolerability supporting the use of 5-ARIs as first-line therapy for men with prostate enlargement.

Further research involving larger patient populations and extended follow-up is warranted to better understand long-term treatment outcomes, including disease progression, requirements for surgical intervention, and the potential impact on patients’ mental health.

Recent studies and scientific publications provide important updates. A population-based analysis demonstrated an association between higher cumulative exposure to 5-ARIs and reduced cardiovascular mortality, while lower exposure was linked to an increased risk of suicidal ideation [24]. Data from the U.S. Food and Drug Administration Adverse Event Reporting System has further confirmed not only the well-recognized sexual side effects of finasteride but also emerging safety signals, including post-5-ARI syndrome and Peyronie’s disease [25]. Importantly, the European Medicines Agency’s Pharmacovigilance Risk Assessment Committee has formally acknowledged suicidal ideation as a rare but genuine adverse effect of finasteride, whereas no comparable safety signal has been identified for dutasteride [26].

This study has several limitations. It was retrospective and based on routine clinical data, which may introduce selection bias and incomplete documentation. The follow-up period of approximately 12 months was relatively short and did not capture long-term outcomes. Some baseline differences between groups, such as prostate volume and PSA levels, could have influenced the results. Several observed differences reached statistical significance but were numerically small and may have limited clinical relevance. In addition, adverse events were not systematically recorded for all patients, which restricts the interpretation of safety data.

Despite these limitations, the study provides useful real-world insights into the comparative effectiveness and tolerability of finasteride and dutasteride in the management of BPH.

These findings strengthen the view that while 5-ARIs remain effective and valuable for the management of BPH, their use requires individualized treatment decisions and ongoing risk assessment, particularly with regard to psychiatric vulnerability and dose-dependent toxicity.

Finasteride: After treatment, correlations between prostate size/PSA and symptoms or flow weaken, showing a partial decoupling of structure from function.

Dutasteride: More correlations remain significant post-treatment, suggesting that structural changes (volume, PVR) continue to track with functional outcomes.

Before treatment: Both groups show broadly similar correlation patterns linking prostate parameters with symptoms and flow.

After treatment: Finasteride emphasizes symptom relief that is less dependent on prostate size, while Dutasteride preserves stronger size–symptom associations.

Overall implication: The two drugs may differ in mechanistic profile—Finasteride acting more on symptom independence, and Dutasteride reflecting both symptom and structural improvement.

## Figures and Tables

**Table 1 medicina-61-01944-t001:** Comparative outcomes of treatment with finasteride and dutasteride after approximately 12 months of therapy in patients with benign prostatic hyperplasia.

Parameter	Finasteride (n = 162) Baseline	Finasteride (After Treatment)	Δ Finasteride	Within-Group *p*	Dutasteride (n = 239) Baseline	Dutasteride (After Treatment)	Δ Dutasteride	Within-Group *p*	Between-Groups *p* (Δ)
IPSS	19.32 ± 2.79	12.53 ± 2.91	−6.79 ± 3.02	<0.0001	18.02 ± 2.68	10.93 ± 2.88	−7.09 ± 3.01	<0.0001	0.2673
QoL	4.29 ± 1.20	2.15 ± 1.27	−2.14 ± 1.60	<0.0001	4.04 ± 1.06	1.69 ± 0.99	−2.35 ± 1.13	<0.0001	0.8557
Prostate volume (mL)	79.46 ± 25.14	67.80 ± 22.64	−11.65 ± 12.71	<0.0001	70.94 ± 15.40	60.73 ± 13.65	−10.21 ± 9.94	<0.0001	0.5863
PSA (ng/mL)	4.58 ± 3.59	3.16 ± 2.06	−1.42 ± 2.29	<0.0001	3.21 ± 1.47	2.29 ± 1.16	−0.92 ± 0.91	<0.0001	0.0443
Qmax (mL/s)	13.81 ± 3.82	15.73 ± 4.05	+1.91 ± 1.83	<0.0001	14.50 ± 4.36	16.63 ± 4.93	+2.13 ± 2.54	<0.0001	0.1135
PVR (mL)	80.59 ± 46.49	59.07 ± 28.96	−21.51 ± 33.43	<0.0001	74.71 ± 39.11	57.06 ± 27.58	−17.65 ± 29.73	<0.0001	0.7140
IIEF-5	11.81 ± 6.43	10.40 ± 6.28	−1.41 ± 2.02	<0.0001	17.00 ± 5.31	14.53 ± 5.02	−2.47 ± 3.20	<0.0001	**0.0015**
Creatinine (mg/dL)	0.95 ± 0.23	0.93 ± 0.21	−0.02 ± 0.16	0.1739	0.86 ± 0.18	0.83 ± 0.15	−0.04 ± 0.15	<0.0001	0.2590

Values are presented as mean ± standard deviation. Δ = change from baseline; within-group comparisons = Wilcoxon test; between-group comparisons = Mann–Whitney U test. IPSS, International Prostate Symptom Score; QoL, quality of life; PSA, prostate-specific antigen; Qmax, maximum urinary flow rate; PVR, post-void residual urine volume; IIEF-5, International Index of Erectile Function-5. After treatment refers to post-treatment values obtained at the 6–12-month visit.

## Data Availability

The original contributions presented in this study are included in the article. Further inquiries can be directed to the corresponding author.

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
