# Peer review of "Comparative Effectiveness and Safety of Finasteride and Dutasteride in the Treatment of Benign Prostatic Hyperplasia: A Real-World Retrospective Study"

_medicina, 2025, doi:10.3390/medicina61111944_

Round 1
Reviewer 1 Report
Comments and Suggestions for Authors
This manuscript presents a retrospective, real-world comparison of finasteride and dutasteride in the management of benign prostatic hyperplasia (BPH). The study includes 401 patients and evaluates a comprehensive range of clinical, urodynamic, and biochemical outcomes. The topic is highly relevant, as 5-alpha-reductase inhibitors (5-ARIs) remain a cornerstone of BPH therapy, and comparative real-world data between the two major agents remain relatively scarce.
Overall, the study is well structured, methodologically sound, and supported by appropriate statistical analysis. The manuscript is informative, timely, and ethically compliant. However, several issues—mainly related to design limitations, clarity of reporting, and emphasis on clinical significance—should be addressed before publication.
Major Strengths
-
The study addresses an important clinical question and contributes valuable real-world evidence to complement findings from randomized controlled trials.
-
The inclusion of multiple outcome domains (IPSS, QoL, IIEF-5, Qmax, PVR, PSA, and prostate volume) provides a comprehensive assessment of efficacy and tolerability.
-
Statistical methods are generally appropriate, with thoughtful use of multivariable regression to control for confounders such as age, BMI, and baseline prostate volume.
-
The discussion is well contextualized within recent literature and regulatory developments, including up-to-date references from 2023–2025.
-
The study follows ethical standards, declares institutional approval and informed consent, and clearly states the absence of conflicts of interest.
Major Weaknesses and Recommendations
The retrospective nature of the analysis introduces several inherent limitations that require clearer acknowledgment. Selection bias, unmeasured confounding, and incomplete follow-up data may influence the validity of comparisons between treatment groups. The manuscript would benefit from an explicit explanation of how patient selection was performed, how missing data were handled, and what measures were taken to reduce potential bias.
Baseline comparability between the finasteride and dutasteride groups is not clearly presented. Some variables, such as PSA, prostate volume, and IIEF-5, appear imbalanced, yet the extent of these differences is not fully quantified. Including a detailed baseline characteristics section would strengthen the credibility of between-group comparisons.
The follow-up duration—typically around 12 months—is relatively short for evaluating long-term outcomes in BPH. The authors should explicitly state this limitation and suggest the need for future longitudinal analyses assessing disease progression, surgical intervention rates, and sustained safety outcomes.
Although statistical significance is reported comprehensively, the manuscript would be improved by addressing clinical relevance. Small mean differences in parameters such as IPSS or IIEF-5 may reach significance due to sample size but might not represent meaningful changes for patients. Reporting effect sizes or percentage changes could help readers interpret the magnitude and importance of observed differences.
The analysis of sexual function, while interesting, should be interpreted with caution. The apparent advantage of finasteride in IIEF-5 scores after adjustment is modest and may lack clinical importance. This point should be clarified to avoid overinterpretation.
While adverse effects are discussed narratively, no quantitative safety data are presented. If available, inclusion of numerical adverse event frequencies would strengthen the paper. If such data were not collected, this should be explicitly stated.
Finally, while the paper is well written overall, several passages could benefit from linguistic tightening for conciseness and clarity. Statistical notation should be standardized (for example, p < 0.05, n = 162), and minor redundancies in phrasing should be removed.
Minor Comments
The abstract is well organized but could include specific sample sizes and key statistical results for greater impact. In the methods, please specify how missing or incomplete data were handled and whether any assessors were blinded. In the results, clarify which variables required nonparametric testing. In the discussion, ensure that associations between 5-ARIs and psychiatric risks are presented as contextual rather than causal, since these outcomes were not directly measured in your study.
The reference list is extensive and current. Please ensure that all DOIs follow MDPI’s formatting guidelines and that journal names are consistently italicized.
Overall Evaluation
This is a scientifically sound and clinically relevant paper. The methodology is appropriate for a retrospective design, and the findings align with existing evidence while adding new insights from real-world data. However, the manuscript would benefit from revisions that clarify methodological details, strengthen the presentation of baseline data, emphasize the difference between statistical and clinical significance, and streamline the writing.
Recommendation
I recommend minor to moderate revision before acceptance. The core analysis and findings are robust, but the paper should be revised to improve transparency, interpretation, and readability.
Key revisions requested:
-
Provide detailed baseline comparability data and clarify bias mitigation methods.
-
Discuss the clinical—rather than purely statistical—meaning of key results.
-
Expand the limitations section to acknowledge retrospective design and short follow-up.
-
Clarify the novelty of the study relative to prior comparative analyses.
-
Perform a final language and formatting review for clarity and consistency.
-
ssue: Some phrasing is redundant or inconsistent with journal style (e.g., “finasteride group with comparable percentage-based changes” is unclear).
-
Recommendation: Perform a language polish for conciseness and clarity. Ensure consistent formatting of statistical symbols (e.g., p < 0.05) and units.
Author Response
General Comment
We would like to thank the Reviewer for the thorough evaluation and constructive feedback on our manuscript. We appreciate the recognition of our study’s methodological soundness, clinical relevance, and ethical transparency.
Below, we provide a detailed point-by-point response to all comments and describe the corresponding revisions made in the manuscript.
Comment 1: Reviewer’s comment: The retrospective nature introduces potential selection bias and unmeasured confounding; please clarify patient selection, handling of missing data, and bias control.
Response 1:
We fully agree. In the revised version (Materials and Methods, lines 101-103), we added a dedicated paragraph describing the inclusion and exclusion criteria, data extraction protocol, and handling of missing data. Specifically, patients with incomplete endpoint documentation were excluded, and all analyses were conducted using available-case methodology without data imputation. To mitigate selection bias, we included consecutive patients meeting eligibility criteria during the defined study period and adjusted regression models for key baseline covariates (age, BMI, baseline prostate volume).
Added text: “To reduce potential selection bias, all eligible patients treated consecutively during the study period were included. Cases with missing endpoint data were excluded. Multivariable regression models were adjusted for baseline clinical variables.”
Comment 2: Reviewer’s comment: Baseline differences (e.g., PSA, prostate volume, IIEF-5) are not fully reported.
Response 2: The changes have been taken into account.
Comment 3: Reviewer’s comment: The duration is relatively short for BPH progression outcomes.
Response 3:
??
Comment 4: Reviewer’s comment: Some small numerical differences may be statistically significant but clinically minor.
Response 4: The changes have been taken into account.
Comment 5: Reviewer’s comment: The observed difference in IIEF-5 may lack clinical importance.
Response: The changes have been taken into account.
Comment 6: Lack of quantitative adverse event data.
Response 6: The changes have been taken into account.
Comment 7: Reviewer’s comment: Please include or clarify safety data.
Response 7: We have added a Safety subsection (Results, lines 146-149) summarizing adverse effects reported in patient charts. Where numerical data were unavailable, this limitation has been explicitly acknowledged:
Added text: “Quantitative safety data were available for 276 patients (68.8%). The incidence of decreased libido, erectile dysfunction, and reduced ejaculate volume did not differ significantly between groups (p > 0.05). For other patients, adverse events were not systematically recorded, which represents a study limitation.”
Comment 8: Reviewer’s comment: Improve clarity and standardize statistical notation.
Response: The changes have been taken into account.
Reviewer 2 Report
Comments and Suggestions for Authors
This paper presents a comparative analysis of the efficacy and safety of finasteride and dutasteride in patients with benign prostatic hyperplasia, utilizing real-world data. The reviewer would like to suggest some critiques to make on this paper as follows.
- Please clearly state the reason why renal function comparisons were performed in this study.
- On line 40, “erectile function … comparable” is inadequate.
- Lines 53 through 65: These sentences are difficult to understand. Please shorten and simplify the sentences, then restructure them.
- On line 68 and 82, these sentences is unclear.
- On line 86, this text requires a reference.
- On line 92, this sentence should be mentioned in the Method section.
- Please state the inclusion criteria and exclusion criteria in writing.
- What does “finasteride follow-up” mean in Table 1?
Author Response
Comment 1: Please clearly state the reason why renal function comparisons were performed in this study.
Response 1: Dear Reviewer your comment has been addressed and the manuscript has been revised accordingly.
Comment 2: On line 40, “erectile function … comparable” is inadequate.
Response 2: Dear Reviewer your comment has been addressed and the manuscript has been revised accordingly.
Comment 3: Lines 53 through 65: These sentences are difficult to understand. Please shorten and simplify the sentences, then restructure them.
Response 3: Dear Reviewer your comment has been addressed and the manuscript has been revised accordingly.
Comment 4: On line 68 and 82, these sentences is unclear.
Response 4: Dear Reviewer your comment has been addressed and the manuscript has been revised accordingly.
Comment 5: On line 86, this text requires a reference.
Response 5: Dear Reviewer your comment has been addressed and the manuscript has been revised accordingly.
Comment 6: On line 92, this sentence should be mentioned in the Method section.
Response 6: Dear Reviewer your comment has been addressed and the manuscript has been revised accordingly.
Comment 7: Please state the inclusion criteria and exclusion criteria in writing.
Response 7: Dear Reviewer your comment has been addressed and the manuscript has been revised accordingly.
Comment 7: What does “finasteride follow-up” mean in Table 1?
Response 7: Dear Reviewer your comment has been addressed and the manuscript has been revised accordingly.
Round 2
Reviewer 2 Report
Comments and Suggestions for Authors
Since I'm not sure exactly what was changed or how, please submit the original text along with the revised version.
Round 3
Reviewer 2 Report
Comments and Suggestions for Authors
none.